# Change in Voice Quality after Radiotherapy for Early Glottic Cancer

**DOI:** 10.3390/cancers14122993

**Published:** 2022-06-17

**Authors:** Jana Mekiš, Primož Strojan, Dušan Mekiš, Irena Hočevar Boltežar

**Affiliations:** 1Department of Anesthesiology, University Medical Centre Maribor, Ljubljanska ulica 5, 2000 Maribor, Slovenia; jana.mekis@ukc-mb.si (J.M.); dusan.mekis@ukc-mb.si (D.M.); 2Institute of Oncology, Zaloška cesta 2, 1000 Ljubljana, Slovenia; pstrojan@onko-i.si; 3Faculty of Medicine, University of Ljubljana, Vrazov trg 2, 1000 Ljubljana, Slovenia; 4Faculty of Medicine, University of Maribor, Taborska ulica 8, 2000 Maribor, Slovenia; 5Department of Otorhinolaryngology and Cervicofacial Surgery, University Medical Centre Ljubljana, Zaloška cesta 2, 1000 Ljubljana, Slovenia

**Keywords:** early glottic cancer, radiotherapy, voice quality, subjective assessment, acoustic analysis

## Abstract

**Simple Summary:**

The voice of a patient is expected to improve after successful radiotherapy for early glottic cancer. The aim of this study was to follow the changes in the voice for two years after a completed treatment. Subjective patient and physician assessments of the voice quality and objective acoustic analyses of the voice of each patient were used to follow up on the voice changes at regular intervals in 50 patients with T1 glottic cancer. A stroboscopy showed the progression of radiotherapy-induced fibrosis of the vocal folds, which influenced their vibration during phonation. A subjective assessment of the voice quality showed a gradual improvement, but an objective measurement showed a deterioration in a few voice parameters. Two years after the treatment, only 16% of patients had normal voices. The main cause of the impaired voice quality was post-radiotherapy scarring of the vocal folds.

**Abstract:**

Our aim was to track the changes in voice quality for two years after radiotherapy (RT) for early glottic cancer. A videoendostroboscopy, subjective patient and phoniatrician voice assessments, a Voice Handicap Index questionnaire, and objective acoustic measurements (F_0,_ jitter, shimmer, maximal phonation time) were performed on 50 patients with T1 glottic carcinomas at 3, 12, and 24 months post-RT. The results were compared between the subsequent assessments, and between the assessments at 3 months and 24 months post-RT. The stroboscopy showed a gradual progression of fibrosis of the vocal folds with a significant difference apparent when the assessments at 3 months and 24 months were compared (*p* < 0.001). Almost all of the subjective assessments of voice quality showed an improvement during the first 2 years, but significant differences were noted at 24 months. Jitter and shimmer deteriorated in the first year after RT with a significant deterioration noticed between the sixth and twelfth months (*p* = 0.048 and *p* = 0.002, respectively). Two years after RT, only 8/50 (16%) patients had normal voices. The main reasons for a decreased voice quality after RT for early glottic cancer were post-RT changes in the larynx. Despite a significant improvement in the voice after RT shown in a few of the evaluation methods, only a minority of the patients had a normal voice two years post-RT.

## 1. Introduction

Nowadays, radiotherapy (RT) and transoral endoscopic laser surgery are considered to be comparable treatment modalities for early (T_1_N_0_M_0_) glottic cancer with regard to the local control of the disease [1,2,3]. Several recent meta-analyses have shown that laser surgery is better in terms of overall survival, laryngeal preservation, and disease-specific survival [3,4]. Although a few studies have reported a comparable voice quality after transoral laser excision and after RT in patients with early glottic cancer [5,6], other studies have reported advantages of RT over laser surgery [7,8,9] and vice-versa [10]. Therefore, a subjective evaluation of the age of the patient, their occupation, their comorbidities, the tumour location, the extent and depth of the invasion, the accessibility of the treatment modalities, and the expertise of the treatment teams as well as the preference of the patient should be among the factors affecting the decision about which treatment modality to use. When the profession of the patient demands a considerable vocal load and good voice quality, RT is generally suggested as the treatment of choice [9,11,12].

In patients with early glottic cancer, the voice quality and voice-related quality of life decrease as the tumour grows [13]. After the completion of RT, the voice improves, but a considerable number of patients do not regain their normal voices [13,14,15]. A normal voice is of paramount importance for everyday communication, particularly for professional voice users. In this group, a good voice quality and an absence of vocal fatigue problems are prerequisites for their return to work [16,17].

There is limited information on the evolution of voice changes after RT for early glottic cancer [7,11,13,18]. These studies include a limited number of patients, various methods for the assessment of the voice quality (subjective and objective), and diverse time periods of follow-up.

In our previous study, we followed the changes in voice quality by means of subjective and objective assessments three months after the completion of RT and noticed an improvement in most of the observed parameters [15]. The aim of the present study was to monitor the evolution of voice quality with serial assessments at predetermined time points until 24 months after the end of RT and to identify the factors affecting voice improvements. We tested the hypothesis that the voice quality would improve at each subsequent follow-up visit. The second hypothesis was that the tumour extent, type of biopsy, degree of radiomucositis, and scarring after radiotherapy would be predictive factors for a poorer voice quality.

## 2. Patients and Methods

The study protocol was approved by the Republic of Slovenia National Medical Ethics Committee (No. 0120-476/2019/7).

### 2.1. Patients

In the prospective study, 77 consecutive patients with histologically confirmed squamous cell carcinomas of the glottis staged as T_1_N_0_M_0_ and treated with curative intent RT at a tertiary centre were included. After the exclusion of 12 patients who withdrew their consent to participate in the study during the post-treatment follow-up, 11 who did not attend all planned follow-up visits, and 4 patients with a local recurrence, the study group consisted of 50 patients (Figure 1).

### 2.2. Radiotherapy

All the patients were immobilised in a supine position using a five-point thermoplastic cast. Computer tomography-based planning with a slice thickness of 3 mm and MV photon beams were employed. The clinical target volume (CTV) included vocal folds with ipsilateral arytenoids, anterior commissure, and the parapharyngeal space. The upper border of the CTV was the most cranial extent of the arytenoid cartilage, superiorly; its lower border was 1–1.5 cm below the level of the true vocal fold, inferiorly. The planning target volume (PTV) was created by adding 5 mm to the CTV in all directions. The prescription dose was 63 Gy (range, 58.5–65.25 Gy) delivered in 2.25 Gy daily fractions over 36 to 49 days (median, 39 days; mean 39.82 ± 2.56 days). A total of 45 patients (90%) received 63 Gy.

### 2.3. Factors Possibly Influencing Voice Quality

The data on gender, age, smoking, the presence of gastroesophageal reflux, allergies, pulmonary diseases, an impaired hearing ability, the type of biopsy (punch biopsy or excisional biopsy), the need for a repeated biopsy, the extent of the glottic cancer (one or both vocal folds), and the degree of radiomucositis at the end of the RT course were obtained from the medical documentation.

### 2.4. Assessment of Laryngeal Function and Post-Radiation Mucosal Changes

All assessments were performed 3 months, 6 months, 12 months, and 24 months after RT.

#### Videoendostroboscopy

In order to evaluate the function of the vocal folds and the post-radiation mucosal changes on them, a videoendostroboscopy was performed at each follow-up. The evaluation was performed at the end of the study by a single expert (IHB) from the recordings of the examinations. The assessment was performed without knowing the name of the patient. The vibration of the vocal folds (with regard to amplitude, regularity, symmetry, and mucosal wave) was assessed as normal or abnormal. Closure of the vocal folds (complete or incomplete) and the mobility of the vocal folds (normal or abnormal, impaired or immobile) were also evaluated. The degree of post-radiation mucosal changes was assessed regarding fibrosis, tissue defects, atrophies, and oedemas of the vocal folds as well as the influence of these changes on the completeness of the closure of the vocal folds, the amplitude of the vibration, and mucosal waves. It ranged from 0 to 3 (0 = no changes, 1 = minor changes, 2 = moderate changes, and 3 = severe changes). In the case that one of the observed parameters of the vibration of the vocal folds (vocal fold closure, mucosal wave, or amplitude of vibration) was impaired, the change was assessed as minor. In the case of two or three impaired parameters, the mucosal alteration was assessed as moderate or severe, respectively. Hyperfunctional voice disorder (HFVD) was diagnosed when an excessive activity of the laryngeal muscles during phonation manifested as a compression of the supraglottic structures.

### 2.5. Assessment of Voice Quality and Its Impact on Quality of Life

#### 2.5.1. Subjective Assessment by Patients

At each study follow-up visit, the patients completed the validated Slovenian translation of the Voice Handicap Index questionnaire (VHI) [19]. As with the original questionnaire [20], it consisted of functional (VHI-f), physical (VHI-p), and emotional (VHI-e) subtests. The patients also assessed their voice quality on a visual analogue scale (VAS, from 0 to 100%) and their vocal fatigue in everyday communication (present or absent).

#### 2.5.2. Subjective Assessment by a Phoniatrician

An auditory-perceptual assessment of the voice quality was performed by a single phoniatrician (IHB) during spontaneous speech of the patient at the time of the follow-up visit prior to performing the videoendostroboscopy. The expert was, therefore, blind to the current status of the larynx. A GRB scoring system was used (G, grade; R, roughness; B, breathiness; graded from 0 to 3 (0 = not present, 1 = minor disorder, 2 = moderate disorder, 3 = severe disorder)) [21].

#### 2.5.3. Objective Acoustic Analysis

A maximal phonation time (MPT) measurement and an acoustic analysis of three vowel/a/samples at the most comfortable pitch and volume were performed for an objective voice quality assessment. The recordings were performed in a room with an environment noise of less than 30 dB using a Shure SM58 microphone (Shure INC, Evanstone, IL, USA). A multi-dimensional voice program (KayPentax^®^, Lincoln Park, NY, USA) was used for a fundamental frequency (F_0_) analysis. F_0_, pitch perturbation (jitter %), and amplitude perturbation (shimmer %) were measured. The mean values of the mentioned parameters and the MPT measurements were calculated for the statistical analysis.

### 2.6. Statistics

The statistical analyses were performed using SPSS version 22.0 (SPSS Inc., Chicago, IL, USA). All statistical tests were two-sided. A statistical significance was set at a *p*-value of <0.05.

The results of the videoendostroboscopy findings (vocal fold vibration, vocal fold closure, post-radiation mucosal changes, signs of HFVD), the auditory-perceptual assessments of the voices of patients, vocal fatigue, the acoustic analysis of the voice samples, the MPT, and the VHI questionnaires were compared between each two consecutive follow-up assessments (at 3 months, 6 months, 12 months, and 24 months after RT) and between the assessments at 3 months and 24 months. The Shapiro–Wilk test was used for the normality testing of the numerical data. A paired *t*-test or non-parametric Wilcoxon signed-rank test was used for a comparison of the data. The categorical variables were presented as frequencies and the differences in the frequency distribution among the different groups were tested by a Fisher exact test or a chi-squared test.

## 3. Results

The study group consisted of 6 women and 44 men aged between 32 and 85 years (mean 62.48 ± 9.99 years). Among them, there were 4 non-smokers, 30 active smokers, and 12 ex-smokers who stopped smoking 6 months before the diagnosis. Information on smoking status was unavailable for four patients. At the end of the two-year follow-up, only two patients were still smoking. Symptoms of gastroesophageal reflux were present in 26 patients. Four patients had a history of allergies, five patients had pulmonary diseases, and seven patients complained of a decreased hearing ability. In order to confirm a malignant disease, a punch biopsy was performed on 13 patients and excisional biopsy on 37 patients. A repeated biopsy because of an inconclusive histopathological diagnosis was necessary for 15 patients. The tumour was limited to the left and right vocal cord (T1A) in 12 and 28 cases, respectively, whereas in 10 cases it involved both vocal cords (T1B). At the end of RT, 34 patients had radiomucositis Grade 2 and 16 patients had Grade 3, according to the National Cancer Institute Common Terminology Criteria for Adverse Events.

Post-radiation mucosal changes in the vocal folds were seen in all but 2 patients at 3 months, in all but 1 patient at 6 months, and in all patients at 12 and 24 months after the RT. Moderate mucosal changes were present in 2 patients at 3 months, in 8 patients at 6 months, in 14 patients at 12 months, and in 18 patients at 24 months after RT. No patients had severe mucosal changes. In four patients, an assessment of the vocal folds was not possible because of compression of the ventricular folds above the vocal folds during the videoendostroboscopy. There were no significant differences regarding the presence and severity (absent or mild vs. moderate) of post-radiation mucosal changes between the consecutive assessments. The only significant deterioration was observed when the results of the assessments at 3 months and at 24 months post-RT were compared (*p* < 0.001) (Table 1).

The vocal fold vibration during the stroboscopy was found to be normal in 8 patients at 3 months, in 9 patients at 6 months, and in 7 patients at 12 and 24 months after RT. There were no significant differences between the consecutive follow-up examinations, nor between the evaluations at 3 months and 24 months.

A complete closure of the vocal folds during vibration was recorded in 30, 36, 36, and 34 patients at 3, 6, 12, and 24 months post-RT, respectively. No significant improvements were detected between the consecutive evaluations, nor between the assessments at 3 and 24 months after RT.

An impaired mobility of one vocal fold was observed during the videoendostroboscopies of three patients at 3 months, in 8 patients at 6 months, and in 8 patients at 12 and 24 months after RT. There were no differences in the comparison between the consecutive evaluations and no significant differences between the examinations at 3 months and 24 months after RT.

There were no significant differences at the consecutive follow-up visits regarding the compression of the ventricular folds as a sign of HFVD. It was noticed in 21 patients at 3 months, in 25 patients at 6 months, in 26 patients at 12 months, and in 27 patients at 24 months after RT. The difference between the 3-month evaluation and 24-month evaluation was also not significant (Table 1).

The auditory-perceptual voice assessment performed by a phoniatrician showed a gradual improvement (decreased value) of two voice parameters, G (grade of hoarseness) and R (roughness), at every consecutive follow-up visit. A statistically significant difference appeared only when the evaluations at 3 months and 24 months post-RT were compared (G: *p* = 0.019; R: *p* = 0.019) (Table 1).

The evaluation of patients of their voice on the VAS scale also showed an improvement at every consecutive follow-up visit. The difference reached a statistical significance between the consecutive visits at 6 months and 12 months after RT (*p* = 0.018). The VAS score at the 24-month visit was significantly higher than at the 3-month visit (*p* = 0.001) (Table 1).

Vocal fatigue was reported by 24, 23, 18, and 18 patients at 3 months, 6 months, 12 months, and 24 months after RT, respectively. There were no significant differences between the consecutive follow-up assessments, nor any differences between the evaluations at 3 months and 24 months after RT.

The results of the VHI with its subtests showed no significant changes at consecutive follow-up visits. However, the difference between the first (3 months after RT) and last evaluation (24 months after RT) was statistically significant for the total VHI score (*p* = 0.007) and physical VHI subtest (*p* = 0.017) (Table 2).

An objective voice evaluation demonstrated no changes in F_0_, but a noticeable deterioration of the stability of the voice pitch (jitter) and amplitude (shimmer) in the first year after RT. No further deterioration was noticed after the first year post-RT. An acoustic analysis of vowel samples showed a significant increase in jitter and shimmer between the sixth and the twelfth month of follow-up (*p* = 0.048 and *p* = 0.012, respectively). When the results of the acoustic analysis between the third and the twenty-fourth month of follow-up were compared, there was also a significant deterioration in jitter (*p* = 0.045) and shimmer (*p* < 0.001) (Table 2).

We tried to identify the patients with normal voices 24 months after RT by taking into consideration the results of several subjective and objective evaluations of the voice quality and their normative data. The auditory-perceptual evaluations of the voice performed by a phoniatrician detected G = 0, R = 0, and B = 0 in 11 patients; the VHI score was below the threshold value of 19.5 [22] in 35 patients and the MPT was above the threshold value of 14 s [23] in 18 patients. Accordingly, only 8 patients (16%) had normal values on all 3 evaluations. These patients had minor mucosal post-radiation changes on the vocal folds and in all of them, the vibration of the vocal folds was normal despite slightly smaller amplitudes; the closure of the vocal folds was complete, but short. The mobility of the vocal folds was normal. However, 3 of these 8 patients reported vocal fatigue after prolonged talking. These three patients demonstrated an approximation of the vocal folds as a sign of a slight hyperfunctional voice disorder on the videostroboscopy.

In order to identify the factors influencing the voice quality after RT, the gender distribution, the mean age, the smoking history, the presence of allergies, pulmonary diseases, gastroesophageal reflux, an impaired hearing ability, the type of biopsy (punch biopsy vs. excisional biopsy), a repeated biopsy necessary for the diagnosis, the tumour extending over both vocal folds, the total radiation dose, the duration of RT, the volume of CTV, the radiomucositis grade at the end of RT, and the degree of mucosal changes on the vocal fold 24 months post-RT were compared between patients with a normal voice and those with a disordered voice. No significant differences between these two groups of patients were found according to the various objective and subjective parameters of the voice evaluation (GRB evaluation, VHI, and MPT). The only parameter that was connected with an abnormal voice was the degree of post-radiation mucosal changes. There were significantly more patients with moderate mucosal changes than in the group with normal voices (18 and 0, respectively, *p* = 0.020). (Table 3)

## 4. Discussion

The results of the present study showed that only a minority of patients with early-stage glottic cancer had a normal voice two years after curative intent RT despite a significant improvement in several voice parameters recorded using the selected evaluation methods. More favourable results were obtained when subjective assessments of the voice quality were used than objective measurements. Post-RT changes in the laryngeal mucosa were identified as the main reason for a decrease in the voice quality.

A combination of subjective and objective evaluation methods as well as an analysis of the consequences of voice variations on the quality of life are recommended for a reliable assessment of the voice quality [24]. Accordingly, in our study, we employed an assortment of subjective and objective methods in order to systematically evaluate the voice quality during the first two years after RT. Previously, we reported short-term results at 3 months post-RT with a significant improvement in voice quality when compared with the voice characteristics before a treatment using the same armoury of subjective and objective evaluation methods [15].

In the present study, we wanted to explore the evolution of the voice quality beyond the third month post-RT in the same group of patients. At 24 months, the results of the VAS assessment of voice quality, total VHI score, VHI-p subscore, and the assessment of voice (G and R parameter) by the phoniatrician were significantly better compared with the assessments at 3 months after the treatment. On the other hand, the degree of mucosal post-radiation changes in the vocal folds, the perturbation of pitch (jitter), and the perturbation of amplitude (shimmer) significantly deteriorated in the same period. Consequently, at 24 months after RT there were only 8 patients (16%) who had normal voices according to the combination of criteria from the different methods employed, including a subjective assessment by the phoniatrician, the VHI results, and the objective measurement of the MPT. In order to identify the normal voices, evaluation methods were chosen that expressed the quality of voice in speech or longer voice samples and not only in short vowel samples. The only factor that significantly influenced the quality of voice 24 months after RT was the degree of mucosal changes observed on the vocal folds. All 8 patients with normal voices had minor mucosal changes. However, although assessed as minor, these changes influenced the vibration of the vocal folds (smaller amplitudes of regular vibrations with a shorter complete closure of the vocal folds) in all these patients, causing an excessive activation of the supraglottic structures during phonation and, consequently, vocal fatigue in three patients.

It is contradictory that the assessment of the phoniatrician, the results of the VHI questionnaire, and the subjective assessment of the voices of patients improved despite a deterioration in the post-radiation mucosal changes seen on the vocal folds. The deterioration in the scarring of the vocal folds was not a surprise. RT affects not only the mucosa, but also the laryngeal muscles, nerves, and vessels in the treated area [25,26]. Histomorphometry, immunohistochemistry, whole-genome microarray, and real-time transcriptional analyses of the irradiated vocal fold tissues showed certain molecular changes (i.e., an increased deposition and disorganisation of collagen, increased fibronectin, and decreased laminin in the thyroarytenoid muscle and/or in the superficial lamina propria), leading to a fibrotic transformation of the vocal folds [27]. In the present study, the post-radiation mucosal changes gradually progressed with no significant deterioration observed between each two consecutive visits; the only significant difference was found when the results of the assessments at 3 months and 24 months after RT were compared. Moreover, there was no significant deterioration regarding an incomplete vocal fold closure, their vibration, and approximation of the ventricular folds as a sign of hyperfunctional disorder. We presumed that the gradual development of the scars enabled the patients to gradually adapt to the new situation and ensure that phonation was as efficient as possible.

An expressed compression of the ventricular fold as a sign of HFVD was noticed in 42% of patients at 3 months and in 54% of patients at 24 months after RT. The aetiology of FVD includes tumour-induced changes in the biomechanics of the vocal cords, radiation-induced fibrosis of the laryngeal structures, and an atrophy of the laryngeal muscles as well as incorrect phonation patterns and excessive talking during the RT course with associated radiomucositis, which also influences phonation. We believe that patients should be well-informed of the need for reduced voice use during the RT course, especially when radiomucositis develops [28,29].

None of the included patients received voice therapy after the treatment completion. Most previous studies have found a positive effect of voice therapy on the voice quality in patients after RT for laryngeal cancer [30,31]. Van Gogh et al. confirmed that the beneficial effect of voice therapy in patients with disordered voices after RT for early glottic cancer may last for a period of at least one year [32]. As an improvement in the voice outcomes found at one-year post-RT was largely maintained in the long term, the role of voice therapy in the early post-RT period and its favourable effect on voice quality should be considered of crucial importance [33]. Thus, it is recommended that voice therapy begins immediately after the resolution of radiomucositis. Moreover, with more sophisticated RT techniques that have an improved control over the dose delivery, additional laryngeal tissue will be spared from higher radiation doses, which will likely further improve the voice quality after RT for early glottic cancer [34]. Therefore, we assumed that, in our patients, voice therapy would reduce excessive laryngeal muscle activity during phonation and consequently vocal fatigue, especially in those with a normal voice quality. In these patients, the vocal cords showed only minor post-RT changes, resulting in more favourable local conditions for regaining a better phonation technique.

In our previous study, we noticed a significant improvement in the subjective voice assessments by a phoniatrician (G and R parameters) and patients (VAS and VHI) and in the objective measurement of the F_0_ and MPT 3 months after RT when compared with the baseline values before the treatment [15]. At the next follow-up visit, we noticed a significant improvement in the VAS, but a deterioration in jitter and shimmer between 6 and 12 months after RT. In the second year after RT completion, there were no significant changes noticed with jitter and shimmer; these showed a non-significant improvement. Watson et al. also noted that there were only minor changes in the voice quality one year after successful RT for early glottic cancer. The authors registered that a high level of voice-related quality of life was maintained many years after curative RT (mean follow-up of 11 years) [33]. Similar results were reported by Naunheim et al. who followed up on patients with early-stage laryngeal cancer from 3 to 20 years after RT. They found no voice deterioration after RT when auditory-perceptual, acoustic, and patient-reported outcomes were taken into consideration [35].

Van Gogh et al. assessed the voice outcome in 106 patients with T1a glottic cancer before and up to 2 years after treatment by an acoustic analysis of voice samples. Two years after RT, only jitter remained significantly different from the normal voices [36]. In our study, only F_0_ returned to the normal values; jitter and shimmer remained outside the normal limits and even deteriorated in the first year after RT.

In the present study, the MPT value at 3 months post-RT did not significantly change (Table 2). We believe that progressive post-radiation changes with an impaired mobility of the vocal folds resulted in an incomplete glottic closure, thus preventing a further improvement in the MPT. Waghmare et al. reported a significant improvement in the MPT in their review of the literature on voice quality after RT for early glottic cancer. They also found out that the voice quality improved after RT although it did not reach the standard of normal voices [37]. The disagreement with our results could originate from the different measurement conditions of both studies.

In our previous study, we noticed a significant improvement in the grade and roughness component of the GRB score recorded by the phoniatrician at three months post-RT [15]. However, significantly different results in the GRB score were documented at 24 months when compared with the baseline or the 3-month evaluation (Table 1). The G and R parameters were normal or only slightly impaired in more than two-thirds of our patients. Marciscano et al. also reported a significant improvement in the voice quality according to the GRBAS score across all evaluated time intervals (more than 12 months post-RT) despite a significant decrease in the motion and vibration of the contralateral vocal cord [18]. Ma et al. also reported on increases in the total GRBAS score several years (mean follow-up, 65 months) after RT [10].

Among our patients, the general VHI score as well as the physical subtest were found to be significantly improved at 24 months post-RT compared with the assessment at 3 months after RT (Table 2). An improvement in the VHI score over time has also been reported by other authors in patients treated with RT for early glottic cancer, but the end results were seldom within normal limits (i.e., below 19.5) [22,38,39,40,41]. In our group, a VHI score below the threshold value was detected in 70% of the included patients at 24 months after RT. The voice quality was also favourably assessed by the patients themselves as the VAS score continuously increased during the 24-month follow-up period (Table 2) with a mean value of 85% at 24 months. A similar pattern of improvement in the VAS score was reported by Bibby et al. [42].

Comparing different voice assessment methods and their results, it appeared that, in our patients, the subjective assessments of voice quality with their consequences on the quality of life were more favourable than the objective measurements. We believe that the phoniatric auditory-perceptual assessment was more complete because the expert routinely assessed the voice of the patient during the conversation before the performing videoendostroboscopy of the larynx. The acoustic analyses and aerodynamic measurements were performed only on short vowel samples, which is not a universal sample of everyday communication. On the other hand, patients assessed the quality of their voice at each follow-up visit. As most of them were smokers, it is possible that their voices were not normal even before the onset of the tumour growth on their vocal folds and they, therefore, assessed their improvement against the previous condition. However, they also completed a VHI questionnaire that showed the handicap on three different domains as a result of the quality of their voice. The results of the VHI questionnaire always reflected the situation at the time of the follow-up visit.

We are aware that our study has certain limitations. A larger number of patients included in the study would certainly increase the statistical power of the results obtained. In this respect, the size of the Slovenian population (about 2 million inhabitants) is a limitation in itself.

There were also limitations to the auditory-perceptual and visual-perceptual assessments. Ideally, videoendostroboscopic recordings and voice samples of the patients should be evaluated by at least two independent evaluators, which would also allow for their inter-rater reliability. Furthermore, there are better auditory analysis protocols for sentence and speech analysis than the F_0_ analysis we used in our study. Regarding the possibility of including a larger number of evaluators and the choice of methods for the acoustic analysis, we were limited when designing the research: the existing protocol of the study reflects the capacities (human and instrumental) that we had at our disposal at the time.

In the auditory-perceptual assessment of voices quality, the GIRBAS scale could be used. This scale also includes an assessment of instability (I), astheny (A), and strained (S) quality of the voice. However, according to the study protocol, only the GRB scale was used at the baseline and 3 months after radiotherapy. Despite a more extensive auditory-perceptual assessment (GIRBAS) of the voices of patients on subsequent follow-up visits, we did not have complete data for all patients. Thus, we only statistically analysed the data for GRB.

## 5. Conclusions

Voice disorders can affect the social and, in several cases, also the professional life of patients and thus impair their quality of life. A decreased voice quality after RT for early glottic cancer is a consequence of post-radiation tissue changes in the larynx leading to the development of FVD in a few patients. The subjective voice assessments provided more favourable results than the objective measurements of voice quality. The subjective parameters improved after RT, but a few objective parameters showed a decrease in the voice quality in the first year after RT. No significant deterioration was noticed later on. Despite the improvement in voice quality recorded post-RT, it was assessed as normal in less than one-fifth of patients. In order to improve an impaired voice quality after RT, voice therapy should be an integral part of the treatment protocols in these patients and offered immediately after the resolution of radiomucositis.

## Figures and Tables

**Figure 1 cancers-14-02993-f001:**
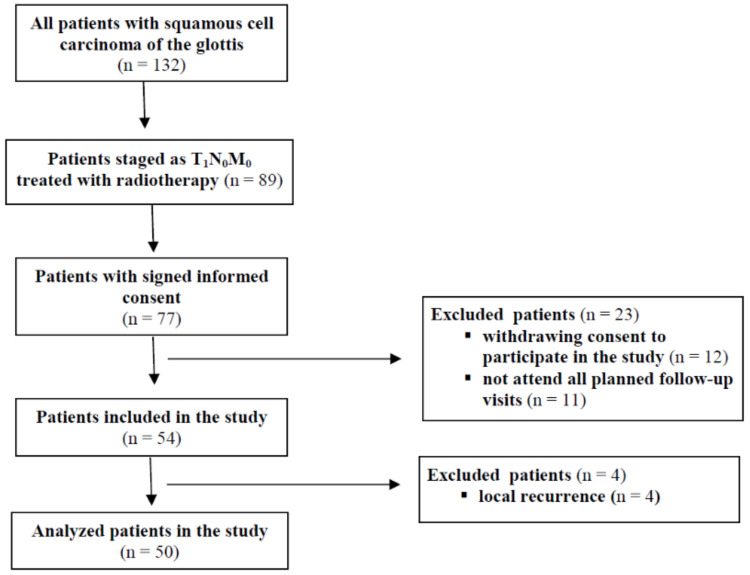
Study flowchart diagram.

**Table 1 cancers-14-02993-t001:** Results of videoendostroboscopies and subjective voice assessments over the 24 months after radiotherapy in patients with T_1_ glottic carcinomas (*N* = 50).

Time of Follow-Up Visit/Comparison Between Follow-Up Visits	Vocal Fold Mucosal Post-Radiation Changes/*p*-Value			HFVD (No)/*p*-Value	GRB Score Mean/SD/*p*-Value			VAS for Voice Assessment (%)/*p*-Value
	PRC Minor or Not Present	PRC Moderate	PRC Severe		G	R	B	
t_3_	42	2	0	21	1.35/0.61	1.28/0.63	0.02/0.15	70.54/24.22
t_6_	41	8	0	25	1.19/0.73	1.17/0.75	0.04/0.20	74.25/22.43
t_3_ vs. t_6_ *p*-value	0.135			0.847	0.070	0.169	0.325	0.678
t_12_	36	14	0	26	1.1/0.62	1.04/0.67	0.02/0.14	83.23/17.46
t_6_ vs. t_12_ *p*-value	0.248			0.917	0.473	0.229	0.569	0.018
t_24_	31	18	0	27	1.04/0.62	0.98/0.73	0.02/0.14	85.07/17.46
t_12_ vs. t_24_ *p*-value	0.475			0.914	0.342	0.252	1.000	0.625
t_3_ vs. t_24_ *p*-value	0.000			0.876	0.019	0.019	1.000	0.001

PRC: post-radiation vocal changes; HFVD: hyperfunctional voice disorder; VAS: visual analogue scale. GRB score: G: grade; R: roughness; B: breathiness; T_3_: time (_months after radiotherapy)_.

**Table 2 cancers-14-02993-t002:** Results of the Voice Handicap Index (VHI) questionnaire, measurement of maximal phonation time, and acoustic analysis of voice samples over the 24 months after radiotherapy in patients with T_1_ glottic carcinomas (*N* = 50).

Time of Follow-Up Visit/Comparison Between Follow-Up Visits	VHI Mean/SD/*p*-Value				MPT (s) Mean/SD/ *p*-Value	Acoustic Analysis Mean/SD/*p*-Value		
	VHI	VHI-f	VHI-p	VHI-e		F_0_ (Hz)	Jitter (%)	Shimmer (%)
t_3_	22.47/23.56	18.98/7.8	11.72/10.25	4.26/7.54	18.98/7.8	152.71/32.46	2.84/4.11	6.64/5.17
t_6_	22.89/26.28	17.29/9.13	12.63/10.64	3.94/7.61	17.29/9.13	153.71/53.61	2.96/4.13	8.9/12.04
t_3_ vs. t_6_ *p*-value	0.686	0.678	0.533	0.161	0.678	0.476	0.575	0.220
t_12_	20.56/26.32	19.37/7.64	10.67/10.37	4.52/8.68	19.37/7.64	157.22/48.55	7.55/14.78	18.55/12.19
t_6_ vs. t_12_ *p*-value	0.635	0.442	0.156	0.512	0.553	0.748	0.048	0.002
t_24_	16.19/23.61	14.91/8.34	8.46/10.11	2.98/6.95	17.79/8.35	154.12/35.4	5.11/7.55	16.82/8.05
t_12_ vs. t_24_ *p*-value	0.454	0.813	0.119	0.596	0.153	0.840	0.235	0.164
t_3_ vs. t_24_ *p*-value	0.007	0.077	0.017	0.053	0.783	0.714	0.045	0.000

T_3_: time _(months after radiotherapy)_; VHI: Voice Handicap Index; F_0_: fundamental frequency; VHI-f: VHI functional; VHI-p: VHI physical; VHI-e: VHI emotional; MPT: maximal phonation time; mean: mean value; SD: standard deviation.

**Table 3 cancers-14-02993-t003:** Comparison between patients with a normal voice (*n* = 8) and patients with a disordered voice (*N* = 42) 24 months after radiotherapy.

Parameter	Patients with a Normal Voice, *N* = 8	Patients with a Disordered Voice, *N* = 42	*p*-Value
Gender (men)	7	37	1.000
Age (years, mean/SD)	60.50/6.72	63.61/9.45	0.381
Tumour extending over both vocal folds	0	8	0.322
Excisional biopsy	6	29	0.663
Repeated biopsy	1	13	0.656
Non-smoker	1	10	0.663
Gastroesophageal reflux	5	21	0.654
Allergy	0	4	1.000
Pulmonary diseases	1	4	0.571
Impaired hearing ability	0	7	0.573
Total RT dose (Gy, mean/SD)	63.28/0.79	63.00/0.87	0.402
Duration of RT (days, mean/SD)	39.25/1.58	39.71/2.33	0.598
3D volume of RT	28.49/7.76	32.81/9.04	0.213
Radiomucositis grade 3	3	13	1.000
Moderate post-radiation mucosal changes on vocal folds	0	18	0.020
Impaired mobility of vocal folds	0	6	0.571
Complete closure between vocal folds	5	30	0.541

## Data Availability

The datasets analysed during the current study are available from the corresponding author on reasonable request.

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
