# Peer review of "Change in Voice Quality after Radiotherapy for Early Glottic Cancer"

_cancers, 2022, doi:10.3390/cancers14122993_

Round 1
Reviewer 1 Report
This paper examines voice quality after radiotherapy for laryngeal cancer. There is no problem with the subject or the method, and the results were interesting. However, I felt that the discussion of the improvement in subjective evaluation despite the poor evaluation of acoustic analysis was a bit incomplete.
Since many of the patients were smokers, it may be that their pre-cancer voice itself was worse than normal in the acoustic analysis. Therefore, the patients themselves may have thought that they were improving.
We can't dismiss the possibility of a bias about the GRB score. I would like to know the number of people who evaluated the GRB score and when they evaluated it (did they evaluate it after knowing the patient's condition or blindly evaluating a recording). (Point 1)
Also, regarding evaluating the mucosa condition by videoendstroboscopy, please describe the evaluation method, such as whether the evaluation was done by several people or one person, whether you evaluated knowing the patient's condition or blindly from the recording. (Point 2)
It would be best to describe the recording environment and equipment regarding acoustic analysis, such as microphones. (Point 3)
I was curious about the change in body weight before and after the treatment, but it is not a problem if there is no data.
In any case, this is a fascinating paper, and I believe that it fully meets the criteria to be accepted by Cancers.
Author Response
Dear Reviewer,
we are grateful for your valuable comments. We tried to meet your suggestions.
- This paper examines voice quality after radiotherapy for laryngeal cancer. There is no problem with the subject or the method, and the results were interesting. However, I felt that the discussion of the improvement in subjective evaluation despite the poor evaluation of acoustic analysis was a bit incomplete.
We thank the reviewer for this comment. We added a sentence that explains a possible cause for the difference between the objective and subjective evaluation of voice quality (lines 384-388).
»We believe that the phoniatric auditory-perceptual assessment was more complete because the expert routinely assessed the patient’s voice during the conversation before performing the videoendostroboscopy of larynx. The acoustic analyses and aerodynamic measurements were performed only on short vowel samples which cannot be a universal sample of everyday communication. «
- Since many of the patients were smokers, it may be that their pre-cancer voice itself was worse than normal in the acoustic analysis. Therefore, the patients themselves may have thought that they were improving.
We thank the reviewer for this valuable comment. We agree that the patients rated their voices based on voice quality before they developed glottic cancer. However, in addition to the subjective assessment on the VAS scale, patients completed a VHI questionnaire in which they had to provide an assessments of different claims relating to physical, functional, and emotional limitations due to voice. These assessments were related to voice quality at the time of evaluation, so we believe that the VHI assessment was less dependent on pre-treatment voice quality. We have inserted the following text to highlight this important issue in the subjective assessment of voice quality (lines 388-394).
»On the other hand, patients assessed the quality of their voice at each follow-up visit. As most of them were smokers, it is possible that their voices were not normal even before the onset of tumor growth on their vocal folds and they therefore assessed improvement against the previous condition. However, they also completed a VHI questionnaire that showed the handicap on three different domains as a result of quality of their voice. The results of the VHI questionnaire always reflected the situation at the time of the follow-up visit. «
- We can't dismiss the possibility of a bias about the GRB score. I would like to know the number of people who evaluated the GRB score and when they evaluated it (did they evaluate it after knowing the patient's condition or blindly evaluating a recording). (Point 1)
We thank the reviewer for a very important question. It was only one expert (a phoniatrician with 20 years of working experience) who evaluated the voice quality at the time of follow-up visits in all patients included in the study. The evaluation was performed before videoendostroboscopy, so the phoniatrician was not yet aware of the patient’s laryngeal condition. Information on this issue has been added as follows (lines 129-134):
»Auditory-perceptual assessment of voice quality was performed by a single phoniatrician (IHB) during the patient’s spontaneous speech at the time of follow-up visit, prior to performing videoendostroboscopy. The expert was therefore blind to the current state of the larynx. The GRB scoring system was used (G- grade, R- roughness, B- breathiness; graded from 0 to 3 (0 = not present, 1=minor disorder, 2=moderate disorder, 3 = severe disorder) [21]«
- Also, regarding evaluating the mucosa condition by videoendstroboscopy, please describe the evaluation method, such as whether the evaluation was done by several people or one person, whether you evaluated knowing the patient's condition or blindly from the recording. (Point 2)
We thank for a valuable comment. We added the missing information about the time and the way of evaluation in the text (lines 104-106 and 114-118).
»The evaluation was performed at the end of the study by a single expert (IHB) from the recordings of the examinations. The assessment was performed without knowing the patient's name. «
»In the case that one of the observed parameters of vocal folds’ vibration (vocal folds’ closure, mucosal wave, amplitude of vibration) was impaired, the change was assessed as minor. In the case of two or three impaired parameters, the mucosal alterations were assessed as moderate or severe, respectively. «
- It would be best to describe the recording environment and equipment regarding acoustic analysis, such as microphones. (Point 3)
We agree with the reviewer that some data on acoustic analysis are missing. We added the necessary information as follows (lines 138-139):
»The recordings were performed in a room with an environment noise of less than 30 dB using Shure SM58 microphone (Shure INC, IL, USA). «
- I was curious about the change in body weight before and after the treatment, but it is not a problem if there is no data.
We are sorry, but we did not record data on patients’ body weight.
- In any case, this is a fascinating paper, and I believe that it fully meets the criteria to be accepted by Cancers.
We are grateful for the encouraging evaluation of our study and manuscript.
Sincerely yours,
Irena Hočevar Boltežar

Reviewer 2 Report
Thank you for the opportunity to provide feedback for the study “Change in voice quality after radiotherapy for early glottic cancer.” The study is the latest manuscript in a series of research on this topic and focus on long-term follow-up through 24 months post radiation. The research can be improved in rigor as follows.
- Please include hypotheses as appropriate
- Include a flowchart for patient recruitment
- Auditory-perceptual and not just perceptual in regard to voice quality ratings because perceptual can, for example, also relate to visual perceptual
- A speech-language pathologist experience in voice disorders should be involved in the auditory-perceptual ratings; best practice would be 2 raters for all visual-perceptual and auditory-perceptual ratings with determination of inter-rater reliability.
- Better to say hyperfunctional voice disorders to be more specific than functional voice disorders, because the interpretation of functional voice disorders can vary and functional voice disorders is now also used to mean an aphonia/dysphonia in the context a functional neurological disorder.
- Objective acoustic measures are not state of the art. Cepstral-spectral measures should be included (for example, cepstral spectral prominence, Praat). That way, sentences and speech can be analyzed as well in addition to vowel prolongation.
- Include a separate limitations section
- Vocal fatigue not voice fatigue
- Vocal folds not vocal cords
Author Response
Dear Reviewer,
we are grateful for your valuable comments. We tried to follow your suggestions.
Thank you for the opportunity to provide feedback for the study “Change in voice quality after radiotherapy for early glottic cancer.” The study is the latest manuscript in a series of research on this topic and focus on long-term follow-up through 24 months post radiation. The research can be improved in rigor as follows.
- Please include hypotheses as appropriate
Thank you for making this very important comment. We tested two hypotheses (lines 67-70)
“We tested the hypothesis that voice quality will improve at each subsequent follow-up visit. The second hypothesis was that tumor extent, type of biopsy, degree of radiomucositis, and scarring after radiotherapy would be predictive factors for poorer voice quality.”
- Include a flowchart for patient recruitment
The flowchart is added and included as Figure 1. Added at the end of the manuscript.
- Auditory-perceptual and not just perceptual in regard to voice quality ratings because perceptual can, for example, also relate to visual perceptual
Thank you for warning. We have changed all the terms in the manuscript accordingly (lines 129, 150, 205, 238, 347).
- A speech-language pathologist experience in voice disorders should be involved in the auditory-perceptual ratings; best practice would be 2 raters for all visual-perceptual and auditory-perceptual ratings with determination of inter-rater reliability.
We agree with this opinion. At the beginning of the study, only one phoniatrician was available to participate in the evaluations. This is the reason why two experts did not participate in the study, which would certainly have been much better. Otherwise, auditory-perceptual rating of voice quality and visual assessment of videoendostroboscopic recordings are among the routine diagnostic procedures performed by a phoniatrician (i.e. ENT specialist with subspecialisation in voice, speech and swallowing), and in our country they are not in the exclusive domain of a speech and language pathologist. We highlighted this problem in the part of the Discussion that refers to the limitations of our study (lines 402-409).
“There are also limitations on auditory-perceptual and visual-perceptual assessments. Ideally, videoendostroboscopic recordings and voice samples of the patients should be evaluated by at least two independent evaluators, which would also allow their inter-rater reliability. Furthermore, there are better auditory analysis protocols for sentence and speech analysis than the F0 analysis we used in our study. Regarding the possibility of including a larger number of evaluators and the choice of methods of acoustic analysis, we were very limited in designing the research: the existing protocol of the study reflects the capacities (human, instrumental) that we had at our disposal at the time.”
- Better to say hyperfunctional voice disorders to be more specific than functional voice disorders, because the interpretation of functional voice disorders can vary and functional voice disorders is now also used to mean an aphonia/dysphonia in the context a functional neurological disorder.
Thank you for this important comment. We have changed all the terms in the manuscript as you suggested (lines 118, 150, 201, Table 1, 216, 247, 309, 312).
- Objective acoustic measures are not state of the art. Cepstral-spectral measures should be included (for example, cepstral spectral prominence, Praat). That way, sentences and speech can be analyzed as well in addition to vowel prolongation.
We agree with this comment. We included in the design of our study those acoustic measurements for which we had the appropriate equipment and experience to work with it. This is why cepstral spectral prominence was not included in the set of measurements. As we followed-up the patients at specific time points and samples of sentences and speech were not recorded, we cannot subsequently add the proposed sentence and speech measurements.
- Include a separate limitations section
Thank you for that suggestion. We have listed the limitations of our study in a separate paragraph, which we added at the end of the Discussion (lines 398-415).
“We are aware that our study has certain limitations. A larger number of patients included in the study would certainly increase the statistical power of the results obtained. In this respect, the size of the Slovenian population (about 2. Million inhabitants) is a limitation in itself.
There are also limitations on auditory-perceptual and visual-perceptual assessments. Ideally, videoendostroboscopic recordings and voice samples of the patients should be evaluated by at least two independent evaluators, which would also allow their inter-rater reliability. Furthermore, there are better auditory analysis protocols for sentence and speech analysis than the F0 analysis we used in our study. Regarding the possibility of including a larger number of evaluators and the choice of methods of acoustic analysis, we were very limited in designing the research: the existing protocol of the study reflects the capacities (human, instrumental) that we had at our disposal at the time.
In the auditory-perceptual assessment of voices quality, the GIRBAS scale could be used. This scale also includes an assessment of instability (I), astheny (A), and strained (S) quality of voice. However, according to the study protocol, only the GRB scale was used at baseline and 3 months after radiotherapy. Despite a more extensive auditory-perceptual assessment (GIRBAS) of patients' voices on subsequent follow-up visits, we do not have complete data for all patients. Thus, we only statistically analyzed the data for GRB.«
- Vocal fatigue not voice fatigue
Changed as indicated (lines 127, 151, 218, 245, 294).
- Vocal folds not vocal cords
Changed as indicated (lines 86, 89).
Sincerely yours,
Irena Hočevar Boltežar

Reviewer 3 Report
I would like to thank the Editor for the invitation to review this paper. The authors present a study about voice quality in patients with early glottic tumors treated with radiotherapy. It is interesting to assess the long-term effects of radiotherapy on the voice of patients, considering the equivalence in oncologic outcomes between radiotherapy and Transoral Laser Microsurgery for these tumors. Although the number of patients is limited, statistically significant results are achieved and highlight the late effects of radiotherapy in the larynx. Nevertheless, some parts of the paper could be improved:
- I miss a more concrete description of the postradiation changes observed on vocal folds.
- GRB can include the most common impaired aspects of the perceptual voice assessment in these patients, but I would prefer the use of GRBAS to make it more comparable to other works about this topic.
Author Response
Dear Reviewer,
we are grateful for your valuable comments. We tried to meet your suggestions.
I would like to thank the Editor for the invitation to review this paper. The authors present a study about voice quality in patients with early glottic tumors treated with radiotherapy. It is interesting to assess the long-term effects of radiotherapy on the voice of patients, considering the equivalence in oncologic outcomes between radiotherapy and Transoral Laser Microsurgery for these tumors. Although the number of patients is limited, statistically significant results are achieved and highlight the late effects of radiotherapy in the larynx. Nevertheless, some parts of the paper could be improved:
- I miss a more concrete description of the postradiation changes observed on vocal folds.
We thank the reviewer for this comment. We supplemented the existing description of post-radiation changes observed on vocal cords (lines 104-106 and 114-118).
»The evaluation was performed at the end of the study by a single expert (IHB) from the recordings of the examinations. The assessment was performed without knowing the patient's name.«
»In the case that one of the observed parameters of vocal folds’ vibrtion (vocal folds ‘closure, mucosal wave, amplitude of vibration) was impaired, the change was assessed as minor. In the case of two or three impaired parameters, the mucosal alterations were assessed as moderate or severe, respectively. «
- GRB can include the most common impaired aspects of the perceptual voice assessment in these patients, but I would prefer the use of GRBAS to make it more comparable to other works about this topic.
We agree with the reviewer that GRBAS is more complete perceptive evaluation of the voice quality than GRB. Unfortunately, at the beginning of the study only the simplified GRB evaluation was used, as proposed by Dejonckere et al, 2001 (Eur Arch Otorhinolaryngol 2001; 258: 77-82.). As we did not have complete GRBAS assessments available in 32% of the evaluations, we decided to include only GBR evaluations. We discuss this problem at the end of Discussion (lines 410-415).
»In auditory-perceptual assessment of the quality of voices, GIRBAS scale can be used. This scale also includes an assessment of the instability (I), astheny (A), and strained (S) quality of voice. However, the protocol of the previous study of voice monitoring of patients with T1 glottis carcinoma before and 3 months after radiotherapy contained only GRB assessment (15), and the presented study was a continuation of the previous one. Despite more extensive auditory-perceptual assessment (GIRBAS) of patients' voices on the subsequent follow-up visits, we did not have complete data for all patients. Thus, only the data for GRB were statistically analyzed. «
Sincerely yours,
Irena Hočevar Boltežar
